# BioSeq-Diabolo: Biological sequence similarity analysis using Diabolo

Hongliang Li[1], Bin Liu[1,2]*

**1** School of Computer Science and Technology, Beijing Institute of Technology, Beijing, China, **2** Advanced Research Institute of Multidisciplinary Science, Beijing Institute of Technology, Beijing, China

* bliu@bliulab.net

**Data Availability Statement:** The datasets used to demonstrate the effectiveness of BioSeq-Diabolo are publicly accessed at http://bliulab.net/BioSeq-Diabolo/download/. The BioSeq-Diabolo web server is freely accessible via http://bliulab.net/BioSeq-Diabolo/, and the source code of stand-

## Abstract

As the key for biological sequence structure and function prediction, disease diagnosis and treatment, biological sequence similarity analysis has attracted more and more attentions. However, the exiting computational methods failed to accurately analyse the biological sequence similarities because of the various data types (DNA, RNA, protein, disease, etc) and their low sequence similarities (remote homology). Therefore, new concepts and techniques are desired to solve this challenging problem. Biological sequences (DNA, RNA and protein sequences) can be considered as the sentences of "the book of life", and their similarities can be considered as the biological language semantics (BLS). In this study, we are seeking the semantics analysis techniques derived from the natural language processing (NLP) to comprehensively and accurately analyse the biological sequence similarities. 27 semantics analysis methods derived from NLP were introduced to analyse biological sequence similarities, bringing new concepts and techniques to biological sequence similarity analysis. Experimental results show that these semantics analysis methods are able to facilitate the development of protein remote homology detection, circRNA-disease associations identification and protein function annotation, achieving better performance than the other state-of-the-art predictors in the related fields. Based on these semantics analysis methods, a platform called BioSeq-Diabolo has been constructed, which is named after a popular traditional sport in China. The users only need to input the embeddings of the biological sequence data. BioSeq-Diabolo will intelligently identify the task, and then accurately analyse the biological sequence similarities based on biological language semantics. BioSeq-Diabolo will integrate different biological sequence similarities in a supervised manner by using Learning to Rank (LTR), and the performance of the constructed methods will be evaluated and analysed so as to recommend the best methods for the users. The web server and stand-alone package of BioSeq-Diabolo can be accessed at http://bliulab.net/BioSeq-Diabolo/server/.

## Author summary

Inspired by the similarities between the biological sequences and human languages, we apply the semantics analysis techniques derived from the natural language processing

alone package of BioSeq-Diabolo can be available at https://github.com/Zimiao1025/Sesica under the BSD-2-Clause license.

**Funding:** This work was supported by the National Natural Science Foundation of China (No. U22A2039, 62271049 and 62250028) to (BL). The funders had no role in study design, data collection and analysis, decision to publish, or preparation of the manuscript.

**Competing interests:** The authors have declared that no competing interests exist.

(NLP) to comprehensively and accurately analyse the biological sequence similarities, and propose a platform called BioSeq-Diabolo for intelligently and automatically analysing biological sequence similarities. BioSeq-Diabolo is an important updated version of Bio-Seq-BLM fo/ccusing on the homogeneous and heterogeneous biological sequence similarity analysing, which is beyond the reach of any exiting software tool or platform. BioSeq-Diabolo is named after a popular traditional sport in China whose components reflect its analysing procedures. When playing diabolo, the diabolo stably spins reflecting that the BioSeq-Diabolo is able to automatically construct the optimized predictors, and analyse the corresponding performance for a specific biological sequence similarity analysis task.

## 1. Introduction

All the information determining the structures and functions of DNA, RNA and protein sequences is in their sequences. As one of the fundamental steps in the biological structure and function studies, analysing biological sequence similarities is the foundation of many tasks in bioinformatics, such as protein remote homology detection [1], protein fold recognition [2], protein structure and function prediction [3–5], non-coding RNA and disease association identification [6,7], etc. Therefore, biological sequence similarity analysis has been attracting more and more attentions.

The biological sequence similarities can be further divided into homogeneous biological sequence similarities and heterogeneous biological sequence similarities according to different data types. For the homogeneous biological sequence similarities, the queries and the retrieved samples are homogeneous. For examples, for protein remote homology detection, both the queries and the retrieved samples are protein sequences. Therefore, the homogeneous biological sequence similarities were applied to detect the remote homology relationships [8]. For the heterogeneous biological sequence similarities, the queries and the retrieved samples are heterogeneous. For examples, for circRNA-disease association identification, the queries are circRNAs, while the retrieved samples are diseases. Therefore, the heterogeneous biological sequence similarities were applied to identify the circRNA-disease associations [9].

Because of its importance, some computational methods have been proposed to calculate the biological sequence similarities. For examples, the methods for calculating homogeneous biological sequence similarities have been proposed based on the alignments in an unsupervised manner, such as PSI-BLAST [10], HHblits [11], HMMER [12], DIAMONDScore [13], etc. Later, the supervised methods were proposed, such as HITS-PR-HHblits [14], SMI-BLAST [15]. These methods have been successfully applied to protein remote homology detection, metagenome analysis, prediction of mammalian N6-methyladenosine sites from mRNAs, etc. Because the queries and retrieved samples are belong to different data types, the alignment methods fail to analyze the heterogeneous biological sequence similarities. In this regard, the supervised methods have been proposed. These methods are based on different features, machine learning techniques, biological sequence association networks, etc. Among these methods, the approaches based on deep learning techniques have achieved the state-of-the-art performance, such as HGATLDA [16], GMNN2CD [17], DeepFRI [4], etc. These methods have been successfully applied to lncRNA–disease association prediction, circRNA–disease associations identification, protein function prediction, etc.

All these aforementioned computational methods for analysing the biological sequence similarities have greatly facilitated the developments of many important tasks in bioinformatics. However, they are still suffering from the following disadvantages: i) It is difficult for the

existing methods to detect the low homogeneous biological sequence similarities among biological sequences sharing remote homology relationships; ii) The accuracy for calculating the heterogeneous biological sequence similarities is relatively low because of the various data types, whose characteristics are hard to be captured and formulated; iii) All these methods are designed for specific tasks, and their performance evaluation for the other related tasks is unexplored. Therefore, their contributions to the related fields are limited.

In this study, we are to propose a platform called BioSeq-Diabolo for analysing biological sequence similarities based on biological language semantics, which is named after a popular traditional sport in China whose components reflect the analysing procedures of BioSeq-Diabolo. When playing diabolo, the diabolo stably spins reflecting that the BioSeq-Diabolo is able to automatically construct the optimized predictors and analyse the corresponding performance for a specific biological sequence similarity analysis task. To the best knowledge of ours, it is the first platform for systematically analysing both the homogeneous and heterogeneous biological sequence similarities. Inspired by the similarities between the biological sequences and the natural languages, we are seeking the semantics analysis techniques derived from the natural language processing (NLP) to comprehensively and accurately calculate the biological sequence similarities. Previous studies have proved that the methods developed for analysing natural languages can be applied to the field of molecular biology based on Noam Chomsky's formal language theory [18]. Hierarchical analogy between natural languages and biological sequences has been carried out by [19], where protein sequences were regarded as the "raw text" carrying high-level "meanings" of the structures and functions of proteins. Following the analogy between biological sequences and natural languages, the techniques derived from NLP have greatly contributed to the development of biological sequence analysis [20]. Furthermore, motivated by language models, the biological language models have been proposed [21], which can be applied to residue-level and sequence-level biological sequence analysis tasks. The protein language model ProtTrans [22] is important for understanding the language of life through self-supervised learning. As shown in **Fig 1**, the biological sequence similarity analysis is particularly similar with the semantics similarity analysis in NLP. Sentences determine their semantics of the corresponding natural language, which are the keys for measuring and judging the similarity between sentences. As the sentences of "the books of life" [18], biological sequences determine their structures and functions, which can be considered as the "semantics" of biological sequences. The computational methods for analysing the semantics of natural languages are mature after being studied for decades, giving us an opportunity to apply these methods for improving the performance of biological sequence similarity analysis. In this regard, a total of 27 methods derived from semantics analysis in NLP have been incorporated into BioSeq-Diabolo to analyse the biological sequence similarities, which capture the relevance of biological sequences from distribution, representation and interaction perspectives. Based on these methods, we can uncover the hidden biological language semantics so as to more accurately analyze the biological sequence similarities. Furthermore, the Learning to Rank (LTR) [8,9,23] was employed to integrate the results of different methods in a supervised manner.

In order to help the researchers to study the computational methods for different aims and tasks, we followed the pipelines of intelligent software tools (BioSeq-Analysis2.0 [24], GAIN [25], BioSeq-BLM [21], iLearnPlus [26], etc) to construct the BioSeq-Diabolo platform. The users only need to input the homogeneous or heterogeneous biological sequence data, BioSeq-Diabolo will automatically construct the computational predictors for biological sequence similarity analysis, evaluate the performance, and analyze the results in different views. BioSeq-BLM [21] is a platform for analyzing biological sequences based on biological language models, which is able to automatically construct computational predictors for classification and

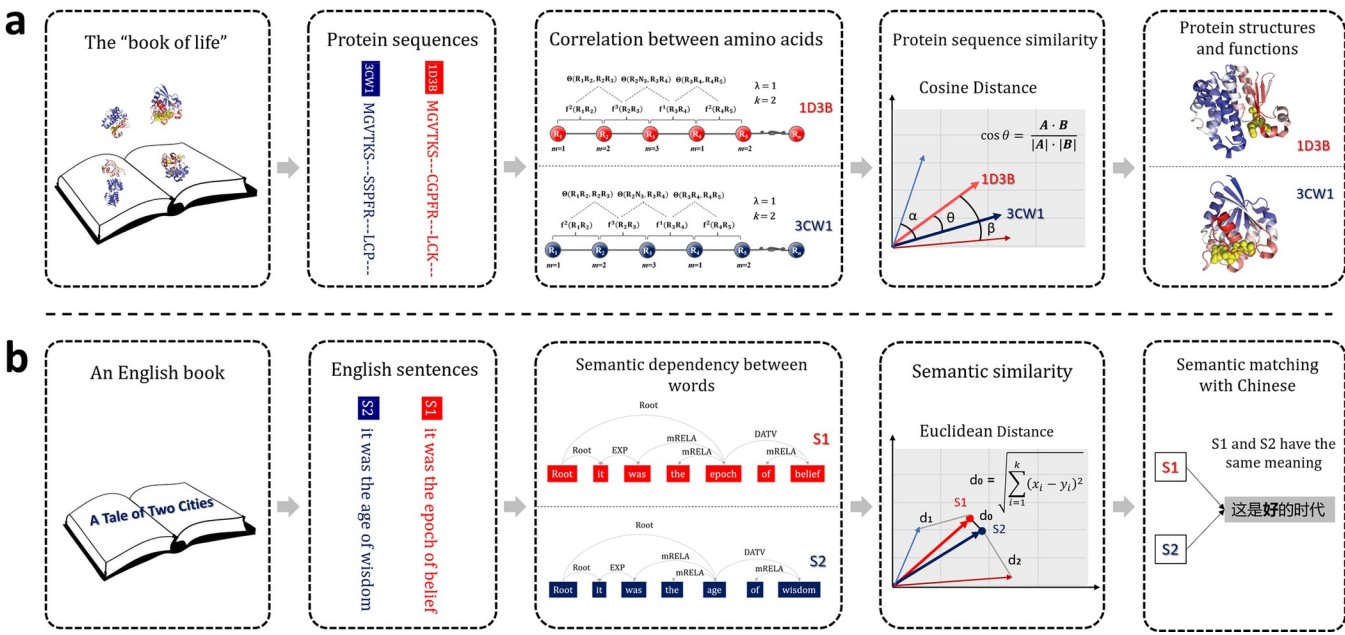

**Fig 1.** The similarities between biological sequence similarity analysis (**a**) and natural language semantics analysis (**b**). Sentences determine their semantics of the corresponding natural language, while biological sequences determine their structures and functions. The biological sequence similarity analysis is particularly similar with the semantics similarity analysis in natural language processing.

sequence labelling tasks. BioSeq-Diabolo is an important updated version of BioSeq-BLM focusing on the homogeneous and heterogeneous biological sequence similarity analysing, which is beyond the reach of any exiting software tool or platform. The comparisons between BioSeq-Diabolo and BioSeq-BLM are listed in **Table 1**. The biological sequence features extracted by BioSeq-BLM can be fed into BioSeq-Diabolo, and the biological sequence similarity scores calculated by BioSeq-Diabolo can also be used as the input features of BioSeq-BLM.

## 2. Materials and methods

### 2.1. Biological sequence similarity analysis tasks

Biological sequence similarities can be further divided into homogeneous biological sequence similarities and heterogeneous biological sequence similarities according to the data types of the target biological sequences. The aim of homogeneous biological sequence similarity analysis is to detect the similarities between two biological sequences with the same data type. For

**Table 1. The comparisons between BioSeq-Diabolo and BioSeq-BLM.**

| Descriptions | BioSeq-BLM | BioSeq-Diabolo |
|---|---|---|
| Biological sequence analysis tasks | Classification and sequence labelling | Sequence similarity analysis |
| Number of classification algorithms | 8 | 0 |
| Number of sequence labelling algorithms | 9 | 0 |
| Number of sequence similarities analysing algorithms | 0 | 27 |
| Categories of algorithms used for result analysis | 4 | 6 |
| Support integration or not | No | Yes |
| Support GPU-accelerate or not | Yes | Yes |

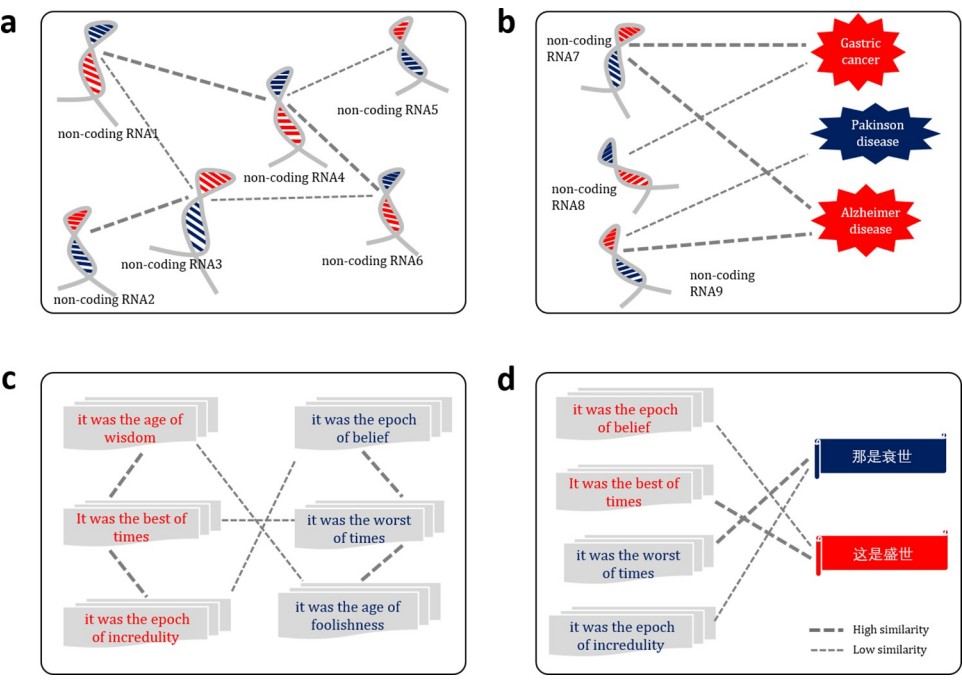

**Fig 2. The analogy between biological sequence similarity analysis tasks and natural language processing tasks.** (**a**) Non-coding RNA similarity analysis, which is a homogeneous biological sequence analysis task. (**b**) Non-coding RNA and disease association identification, which is a heterogeneous biological sequence analysis task. (**c**) Text matching task, which is a homogeneous language analysis task. (**d**) Machine translation task, which is a heterogeneous language analysis task. Thick dotted lines indicate high similarity, and thin dotted lines indicate low similarity. The same shape represents homogeneous association, while different shapes represent heterogeneous association.

examples, **Fig 2A** shows the process of analysing the RNA sequence similarity, which is a homogeneous biological sequence similarity analysis task. **Fig 2B** shows the process of identifying the associations between non-coding RNAs and diseases, which is a heterogeneous biological sequence similarity analysis task.

## 2.2. Biological sequence similarity analysis based on biological language semantics

Semantics represent the meanings of sentences in a particular context. Even the same sentence in different contexts would have different semantics. In order to understand the semantics, the sentences are represented as abstract representation considering both local and global information of the context, and then the advanced natural language processing techniques are performed on the abstract representation to automatically understand the semantics. The semantics analysis plays a key role in NLP, which is important for machine translation, information retrieval, text generation, etc. As the language of "the books of life", biological sequences can be considered as the sentences containing all the information for determining their semantics (the structures and functions of the biological sequences). Therefore, the ideas and techniques derived from semantics analysis can be applied to analyse the "semantics" of biological sequences (see **Figs 1**, **2C and 2D**). 27 different biological sequence similarity analysis methods were incorporated into BioSeq-Diabolo to analyse the biological sequence similarities. These methods can be divided into 3 categories, including distribution methods, representation methods and interaction methods. The following sections will introduce these 27 methods and their embeddings.

## Embeddings

Embeddings transfer biological sequences to dense vectors via incorporating local and global information (**Fig 3A**). Based on the effective embeddings, the biological sequence similarity analysis methods can capture hidden patterns from biological sequences, which is fundamental for learning biological language semantics. Intelligent platforms are able to generate different embeddings, such as BioSeq-Analysis2.0 [24], BioSeq-BLM [21], iLearnPlus [26], etc. These platforms and tools represent the biological sequences based on different techniques and theories.

## Distribution methods

Distribution methods fully consider the spatial correlation of input pairs, and show good generalization ability for modelling different types of data [27]. After encoded by the embedding layers (**Fig 3A**), the biological sequences are projected into the high-dimensional space by the mapping layer, and different algorithms are performed on this high dimensional space to calculate the probabilities, treated as the similarity scores (**Fig 3B**). There are 9 models in distribution methods. For more detailed information of the distribution methods, please refer to **S1 Table**.

## Representation methods

The representation methods employ the Siamese architecture to encode the sentences [27], which can be applied to analyse the biological sequence similarities. Based on Siamese architecture in Contrastive Learning, representation methods efficiently learn the hidden difference and connection of homogeneous biological sequences. Representation methods apply the symmetrical representation layers to extract biological language semantics and matching layers to conduct semantic matching operations based on metric learning (**Fig 3C**). After semantic matching, the similarity scores are generated by calculating the posteriori probabilities. There

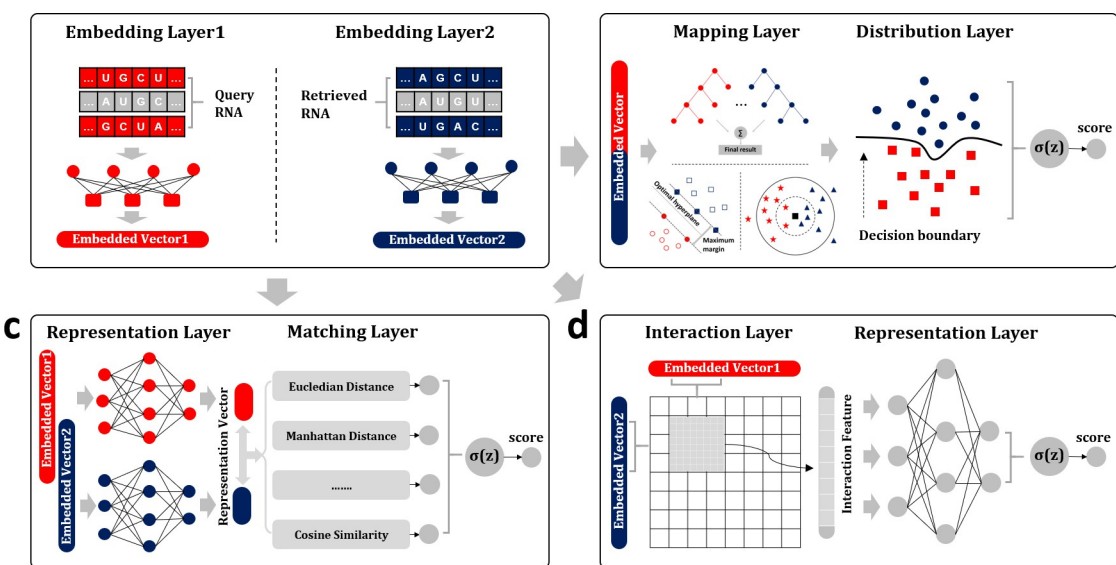

**Fig 3. The architectures of biological sequence similarity analysis methods based on biological language semantics.** (**a**) The architectures of embeddings. (**b**) The architectures of distribution methods. (**c**) The architectures of representation methods. (**d**) The architectures of interaction methods.

are 4 models in representation methods. For more information of the representation methods, please refer to **S2 Table**.

## Interaction methods

Interaction methods employ the hierarchical deep architecture to learn the semantics from the local interaction matrix of query and retrieved documents [27], which is suitable for comprehensively learning the associations between biological sequences. Based on biological sequences and their embeddings, the relevance of two biological sequences is captured by the interaction layer in interaction methods, and the following representation layers learn biological language semantics from interaction matrix (**Fig 3D**). The similarity scores are generated by calculating the posteriori probabilities. There are 14 models in interaction methods. For more information of interaction methods, please refer to **S3 Table**.

## 2.3. BioSeq-Diabolo

Because the biological sequence similarity analysis tasks are diverse with different features and input data, it is often very difficult for the researchers to select the suitable computational methods and techniques for their own tasks and aims. In this regard, an intelligent platform which can automatically construct computational methods for biological sequence similarity analysis, and select the optimized methods for specific tasks is highly required. In this study, we construct a powerful platform called BioSeq-Diabolo for automatically analysing biological sequence similarities based on biological language semantic (**Fig 4**). The users only need to input the biological sequence data and the parameters, BioSeq-Diabolo will intelligently identify the specific tasks (homogeneous or heterogeneous biological sequence similarity analysis tasks) based on the input embeddings of the biological sequences and the parameters provided by the users, and then will accurately analyse the biological sequence similarities based on biological language semantics. BioSeq-Diabolo will integrate the biological sequence similarities

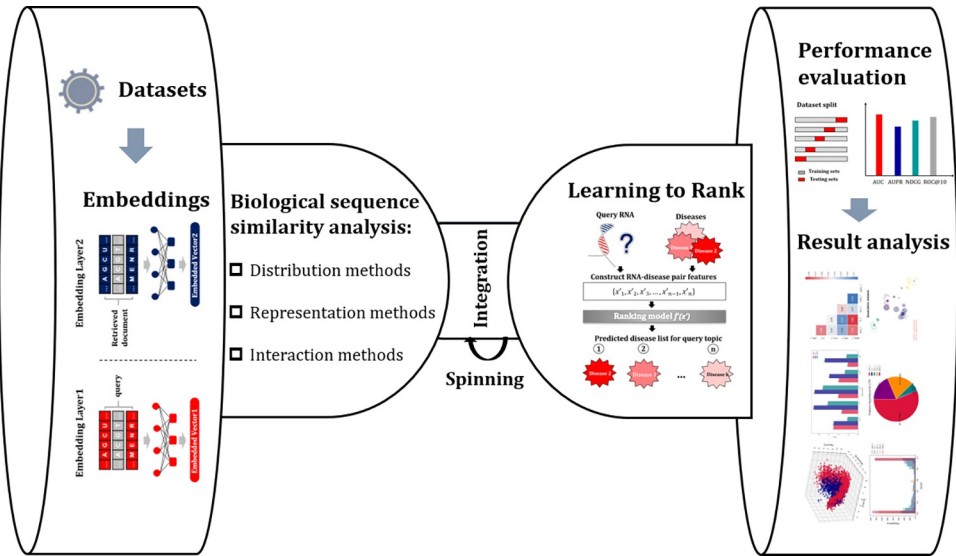

**Fig 4. BioSeq-Diabolo schematic overview.** BioSeq-Diabolo is named after a popular traditional sport in China whose components reflect its analysing procedures. When playing diabolo, the diabolo stably spins reflecting that the BioSeq-Diabolo is able to automatically construct the optimized predictors, and analyse the corresponding performance for a specific biological sequence similarity analysis task.

in a supervised fashion by using Learning to Rank (LTR) [28]. Finally, the performance of different constructed methods will be evaluated and analysed.

## Learning to rank

There are 27 biological sequence similarity analysis methods in BioSeq-Diabolo, leading to $1.09 \diamond 10^{28}$ (27!) different combinations. Furthermore, many embeddings are proposed. For example, BioSeq-BLM [21] can generate 155 different embeddings of biological sequences based on different biological language models. Therefore, the number of base models generated by BioSeq-Diabolo via different biological sequence similarity analysis methods and embeddings is even unlimited. To fully consider the advantages of these base models, we employed Learning to Rank (LTR) [29] to integrate these methods following ProtDec-LTR [8], GOLabeler [23], iCircDA-LTR [9] and DrugE-Rank [30]. Learning to Rank ranks the relevance and importance between queries and documents in a supervised manner. With its help, the biological language semantics will be efficiently explored and uncovered. For more information of LTR, please refer to **S1 Text**.

## Performance evaluation

Four performance measures were used to evaluate the performance of different methods generated by BioSeq-Diabolo, including area under the Receiver Operating Characteristic Curve (AUC) [31], area under the Precision-Recall Curve (AUPR) [32], Normalized Discounted Cumulative Gain (NDCG) [33], ROC [34], $F_{max}$ [35] and $S_{min}$ [35]. Please refer to **S2 Text** for details.

## Result analysis

The result analysis module in BioSeq-Diabolo provides multiple visualization functions for output results so as to intuitively show the performance of the predictors. The result analysis module mainly includes the following visualization functions: 1) Performance visualization. BioSeq-Diabolo provides Receiver Operating Characteristic (ROC) curve, Precision-Recall (PR) curve, Radar map and histogram to visualize the performance of the constructed predictors. 2) Similarity network. BioSeq-Diabolo uses NetworkX [36] library to draw the similarity network according to the predictions of the best generated predictors. 3) Complementarity and relevance map. The Heatmap is automatically generated to show the Pearson Correlation [37] of similarity scores predicted by generated predictors. 4) Score distribution. The distribution histogram for similarity scores of constructed predictors is drawn to compare their interval differences. 5) Contribution of integrated predictors. The Learning to Rank calculates the contribution weights of integrated predictors, and the pie chart is drawn to show the contribution of each predictor. 6) Comparison between embeddings and similarity scores. T-distributed Stochastic Neighbor Embedding (TSNE) [38] is applied to show the difference between input embeddings and similarity scores based on dimension reduction.

To summarize, BioSeq-Diabolo constructs various predictors based on biological language semantics, and integrates them to generate the best predictor for specific biological sequence similarity analysis task. The performance of the predictor is then evaluated, and the visualization of the prediction results improves the interpretability of the constructed predictors. All these complicated processes will be automatically conducted by using BioSeq-Diabolo by using only one command line.

### 2.4. Web server and stand-alone package of BioSeq-Diabolo

Based on the flowchart of BioSeq-Diabolo (S1 Fig), the web server and stand-alone package of BioSeq-Diabolo were developed to facilitate the researchers for analyzing the biological sequence similarities based on biological language semantics.

### Web server

We provide detailed tutorial and documents to explain each procedure and option of the web server, which can be accessed at http://bliulab.net/BioSeq-Diabolo/. More specifically, after selecting biological sequence similarity analysis task, choosing biological sequence similarity calculation methods and setting using Learning to Rank or not, the users will see the input page (see http://bliulab.net/BioSeq-Diabolo/server/). The users need to select the parameters for each module in this page, and input biological sequence data in the BLS format. Please refer to the document of BioSeq-Diabolo for detailed descriptions of BLS format. After submitting the form of the input page, the web server will conduct the calculation according to the pipelines. When the calculation is complete, the results will be displayed in the result page (see http://bliulab.net/BioSeq-Diabolo/server/graph_ssc/rank/submit/result/user). The result page includes visualization results, downloadable files and the similarity scores predicted by Bio-Seq-Diabolo. The command lines of stand-alone package are also provided in the result page so as to help the users to perform the analysis by using their own computing resources via stand-alone package of BioSeq-Diabolo.

### Stand-alone package

Stand-alone package ensures that users can make full use of their own computing resources. There are four modules in the BioSeq-Diabolo stand-alone package: 1) biological sequence similarity analysis: 'sesica_arc.py', 'sesica_clf.py', scripts in 'arc' folder and scripts in 'clf' folder; 2) Learning to Rank: 'sesica_rank.py' and scripts in 'rank' folder; 3) Performance evaluation: scripts in 'utils' folder; 4) Result analysis: 'sesica_plot.py' and scripts in 'plot' folder. Additionally, the multiprocessing technique and GPU acceleration are employed to improve its execution efficiency. The Scikit-learn [39] and MatchZoo [40] python libraries are used to implement the stand-alone package. As discussed in results and discussion section, compiled biological sequence similarity analysis tasks can be easily solved by only using one command line with the help of the stand-alone package of BioSeq-Diabolo. Please refer to the README file in the stand-alone package for more details of the three examples shown in results and discussion section.

## 3. Results and discussion

BioSeq-Diabolo is able to facilitate the development of the computational methods for biological sequence similarity analysis. In this section, we will show the effects of BioSeq-Diabolo for automatically developing computational predictors for solving three important biological sequence similarity analysis tasks, including one homogeneous biological sequence similarity analysis task (protein remote homology detection) and two heterogeneous biological sequence similarity analysis tasks (circRNA-disease associations and protein function annotation). For each task, BioSeq-Diabolo automatically constructs various computational predictors, and selects the best one for the following analysis (see the README file in the stand-alone package of BioSeq-Diabolo).

## 3.1. BioSeq-Diabolo facilitates the protein remote homology detection

Protein remote homology detection is one of fundamental research tasks in protein sequence analysis, which has been extensively studied for decades [1]. However, because proteins with remote homology relationship share very low sequence similarities (<30%), the existing computational predictors fail to accurately detect the protein remote homologous. Therefore, new concepts and techniques are desired to solve this challenging problem. Here, we investigate if BioSeq-Diabolo can construct powerful computational predictors for protein remote homology detection or not. Trained and evaluated on a benchmark dataset constructed based on SCOP1.75 database [41] (http://bliulab.net/BioSeq-Diabolo/download/), BioSeq-Diabolo constructs various predictors, and analyses their performance by using the following command line:

*python sesica_arc.py -data_type homo -bmk_vec bmk_vec.txt -bmk_label pos_label.txt neg_label.txt -arc dssm cdssm drmm drmmtks match_lstm duet knrm -metric roc@50*

The performance analysis results of the top 5 best predictors constructed by BioSeq-Diabolo are shown in **Fig 5**, from which we can see that these predictors perform well for protein remote homology detection, and they are complementary. Is it possible to combine these complementary methods to further improve the predictive performance? In order to answer this question, these methods are combined by a supervised framework Learning to Rank (LTR) by using BioSeq-Diabolo, and the corresponding results are shown in **Table 2** along with the performance of the other state-of-the-art predictors in this field, including PSI-BLAST [10], DELTA-BLAST [42], HHblits [11] and HHsearch [43]. From this table we can see that the predictor automatically constructed by BioSeq-Diabolo outperforms all the other approaches

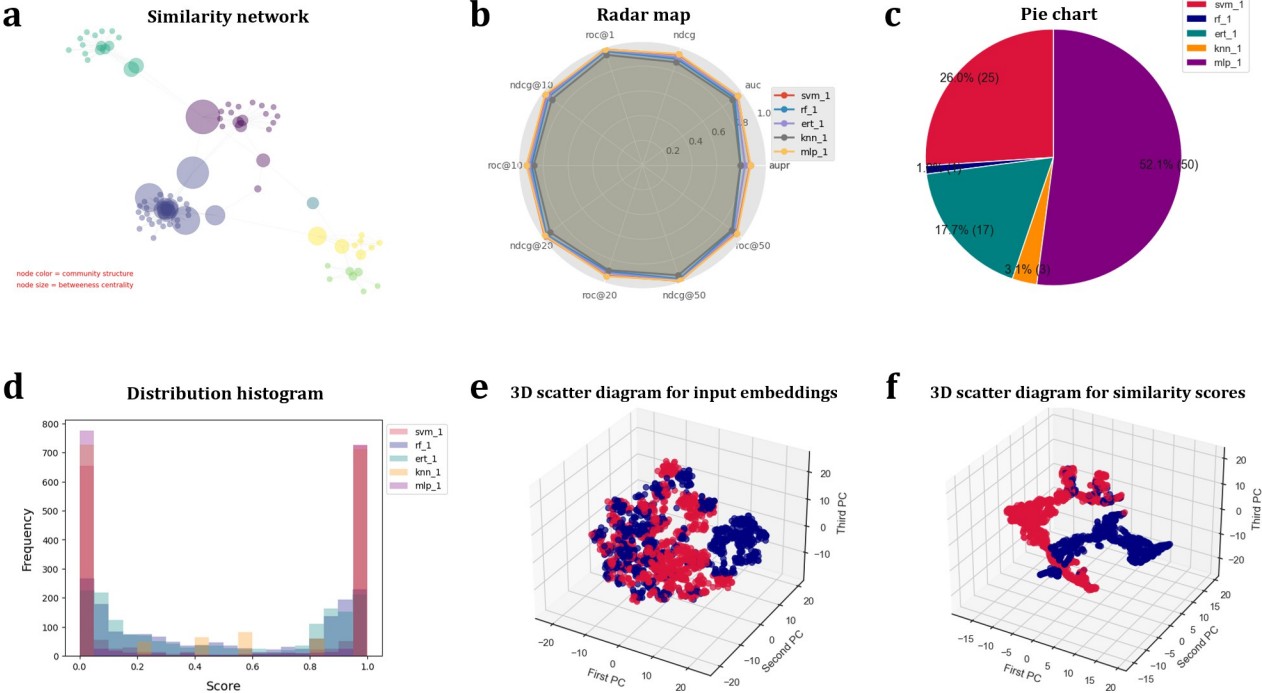

**Fig 5. The visualization results automatically generated by BioSeq-Diabolo for protein remote homology detection.** (**a**) Similarity network predicted by the best predictor constructed by BioSeq-Diabolo. (**b**) Radar map for performance comparison among the top 5 best predictors. (**c**) Pie chart for contribution weights of the integrated top 5 best predictors calculated by Learning to Rank. (**d**) Distribution histogram for similarity scores of the top 5 best predictors. (**e**) 3D scatter diagram for dimension reduction of input embeddings. (**f**) 3D scatter diagram for dimension reduction of similarity scores.

**Table 2. The performance of BioSeq-Diabolo compared with competing methods for protein remote homology detection.**

| Methods | ROC50[c] |
|---|---|
| PSI-BLAST[a] | 87.40% |
| DELTA-BLAST[a] | 88.49% |
| HHblits[a] | 90.52% |
| HHsearch[a] | 90.45% |
| BioSeq-Diabolo[b] | **92.00%** |

[a] The results of the competing methods were obtained from [45]. These competing methods and BioSeq-Diabolo were evaluated on the same test dataset. Therefore, these results can be directly compared

[b] The results of the best predictor constructed by BioSeq-Diabolo (integrating the top 5 best predictors by using Learning to Rank). The input protein sequence embeddings are based on 2-Kmer [46] by using BioSeq-BLM [21]

[c] The performance evaluation indicators were described in **S2 Text** and details of the reported experiments were described in **S3** and **S4**Texts.

in terms of ROC50. These results are not surprising because the semantics analysis techniques derived from natural language processing incorporated in BioSeq-Diabolo are able to efficiently capture the insightful semantics of protein sequences, which are critical for protein remote homology detection [44].

Estimation of time consumption is important for a user-oriented method. Therefore, we evaluated the running time of BioSeq-Diabolo for protein remote homology detection on the proteins randomly extracted from SCOP1.75 database [41] (http://bliulab.net/BioSeq-Diabolo/download/) with the following command line:

```
python sesica_clf.py -data_type homo -bmk_vec bmk_vec.txt -bmk_label pos_label.txt neg_label.txt -clf svm rf ert knn mlp -metric roc@1
```

With the increase of the number of samples, the training time of BioSeq-Diabolo increases obviously, while its test time is roughly linearly with the number of test samples (see **Fig 6**). BioSeq-Diabolo achieved an ROC1 of 0.927 trained with 50000 samples for predicting 50000 test samples. The corresponding training time and test time are 3580 seconds and 243 seconds, respectively. For large scale analysis, we suggest the users to use the stand-alone package. And when the number of samples is the same, the time required for different methods to get the result from scratch are show in **S2 Fig**. Interaction methods and representation methods (both are based on deep-learning) consume more time than distribution methods (based on traditional machine learning methods).

## 3.2. BioSeq-Diabolo facilitates the circRNA-disease association identification

CircRNAs are major regulators in various cellular processes, and associated with the pathogenesis of human diseases. Exactly identifying circRNA-disease associations is critical for researching disease mechanism, and developing corresponding drug targets. Because the existing computational methods ignore the fact that there is a high false positive rate in the negative set, most of existing computational predictors fail to detect potential missing relationship between circRNAs and diseases [9]. Therefore, new approaches for detecting the diseases associated with circRNAs are urgently needed. Here, we applied BioSeq-Diabolo to improve the performance of circRNA-disease association identification. Trained and evaluated on the

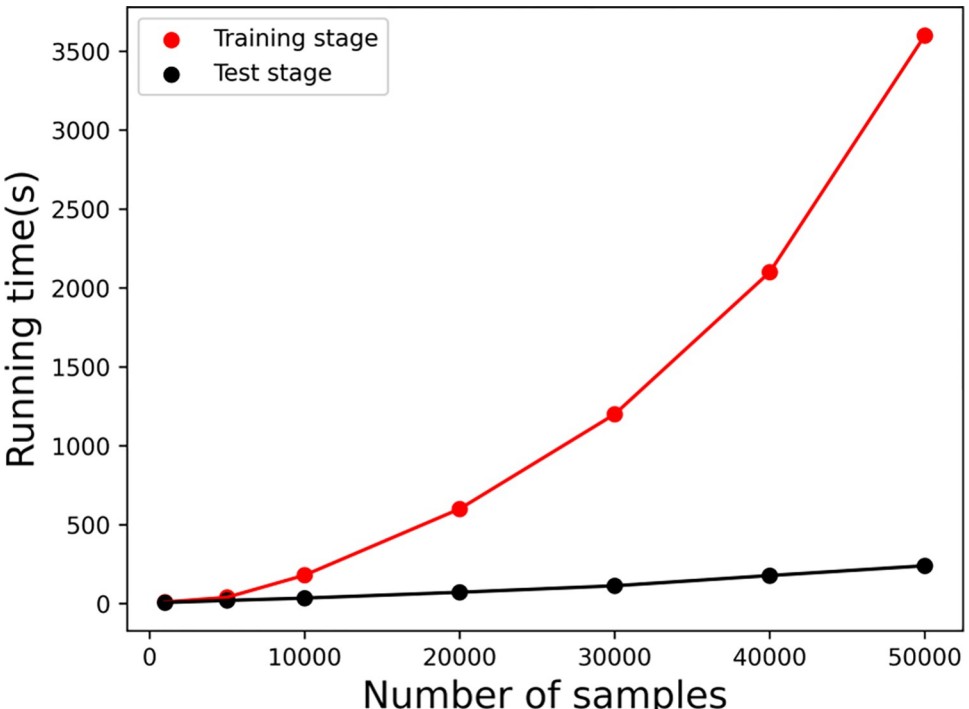

**Fig 6. The training time and test time of BioSeq-Diabolo trained and tested with different number of samples.**
The test time was evaluated with the corresponding BioSeq-Diabolo trained with the same number of samples. These
experiments were performed on Intel(R) Xeon(R) CPU E5-2660 v3 (2.60 GHz with 10 cores) and memory of 64 G.

benchmark dataset [9], BioSeq-Diabolo constructs numerous predictors, and selects the best
predictor by using the following command line:

```
python sesica_clf.py -data_type hetero -bmk_vec_a bmk_circRNA.txt -bmk_vec_b bmk_disease.
txt -bmk_label pos_label.txt neg_label.txt -clf svm rf ert knn mnb gbdt goss dart mlp -metric auc
-gs_mode 2
```

The performance analysis results of the top 5 best predictors constructed by BioSeq-Diabolo
are shown in **Fig 7**, indicating that these predictors can accurately identify circRNA-disease
associations, and they are complementary. In this regard, we employ Learning to Rank to integrate these 5 complementary predictors, and construct the best predictor by using BioSeq-Diabolo. The performance of the best predictor constructed by BioSeq-Diabolo is shown in
**Table 3** along with the performance of the other state-of-the-art predictors in this field, including gcForest [47], DWNN-RLS [48], GBDT [49] and iCircDA-LTR [9]. From this table we can
see that the predictor automatically constructed by BioSeq-Diabolo outperforms all the other
approaches in terms of NDCG and NDCG@10. Based on biological language semantics and
Learning to Rank, BioSeq-Diabolo takes the global relationships among circRNAs and all candidate diseases into consideration, which is the main reason for its better performance.

### 3.3. BioSeq-Diabolo facilitates the protein function annotation

Because only a few proteins have experimentally validated functional annotations, the computational annotations of protein functions have become a crucial step for understanding of the

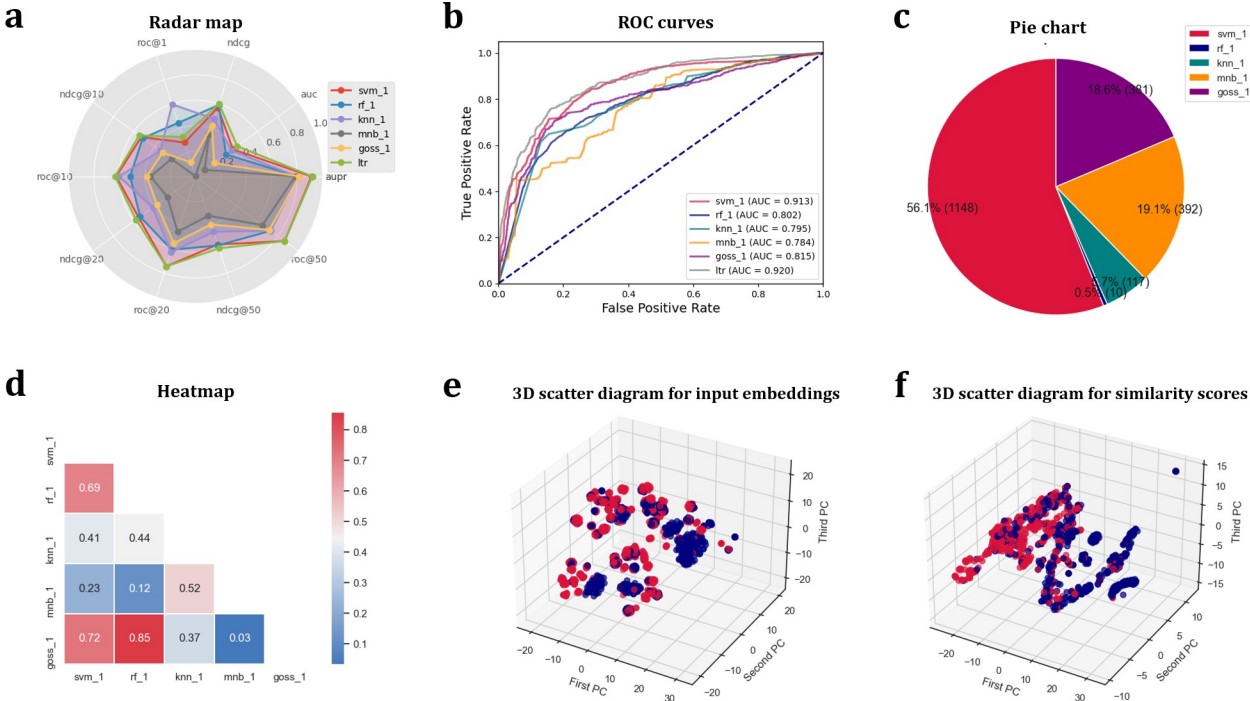

**Fig 7. The visualization results automatically generated by BioSeq-Diabolo for circRNA-disease association identification.** (**a**) Radar map for evaluating performance of the top 5 best predictors. (**b**) ROC curves of the top 5 best predictors. (**c**) Pie chart for contribution weights of integrated top 5 best predictors calculated by Learning to Rank. (**d**) Heatmap for complementarity and relevance of the top 5 best predictors. (**e**) 3D scatter diagram for dimension reduction of input embeddings. (**f**) 3D scatter diagram for dimension reduction of similarity scores.

complex mechanisms of living cells [51]. Although some computational predictors have been proposed to predict protein functions, their performance is still limited prevented by the undefined relationships between protein sequences and their multiple hierarchically organized labels [52]. BioSeq-Diabolo integrates biological sequence semantics from different perspectives, facilitating protein function prediction. Trained and evaluated on a benchmark dataset constructed based on CAFA3 database [52] (http://bliulab.net/BioSeq-Diabolo/download/),

**Table 3. The performance of BioSeq-Diabolo compared with competing methods for circRNA-disease association identification.**

| Methods | NDCG[c] | NDCG@10[c] |
|---|---|---|
| gcForest[a] | 0.4406 | 0.3767 |
| DWNN-RLS[a] | 0.5466 | 0.4911 |
| GBDT[a] | 0.5736 | 0.5280 |
| iCircDA-LTR[a] | 0.5879 | 0.5426 |
| BioSeq-Diabolo[b] | **0.6009** | **0.5492** |

[a] The results of the competing methods were obtained from [9]. These competing methods and BioSeq-Diabolo were evaluated on the same test dataset. Therefore, these results can be directly compared

[b] The results of the best predictor constructed by BioSeq-Diabolo (integrating the top 5 best predictors by using Learning to Rank). The input CircRNA embeddings are extracted by BioSeq-Analysis2.0 [24] with PseKNC [50] (parameter φ is set as 0.5). The input disease embeddings are represented by semantic similarity score matrix reported in [9]. We concatenated circRNA features and disease features, fed into BioSeq-Diabolo for further analysis.

[c] The performance evaluation indicators were described in S2 Text and details of the reported experiments were described in S3 and S4 Texts.

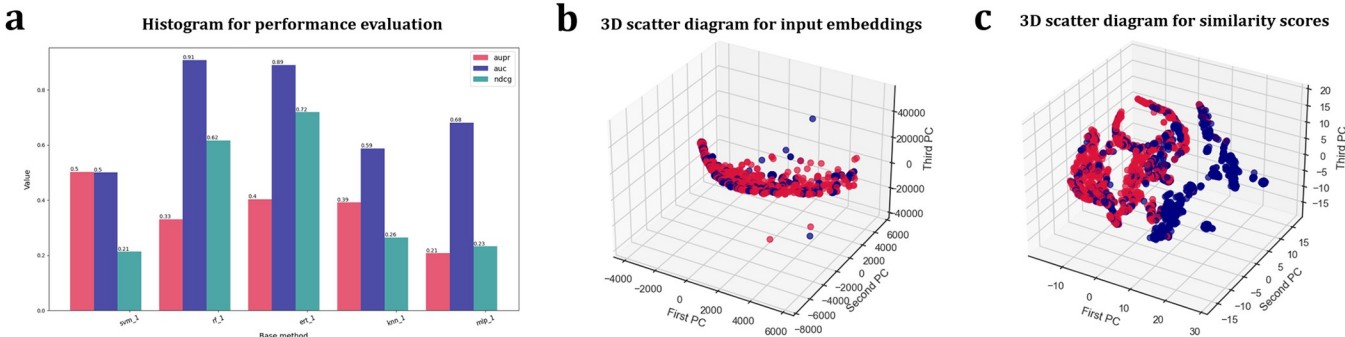

**Fig 8. The visualization results automatically generated by BioSeq-Diabolo for protein function annotation.** (a) Histograms for evaluating performance of the top 5 best predictors. (b) 3D scatter diagram for dimension reduction of input embeddings. (c) 3D scatter diagram for dimension reduction of similarity scores.

BioSeq-Diabolo constructs various predictors, and analyses their performance by using the following command line:

```
python sesica_clf.py -data_type hetero -bmk_vec_a cc_bmk_vec_a.txt -bmk_vec_b
cc_bmk_vec_b.txt -bmk_label pos_label.txt neg_label.txt -clf svm rf ert knn mnb gbdt goss dart
mlp -metric aupr -gs_mode 2
```

The performance analysis results of the top 5 best predictors constructed by BioSeq-Diabolo are shown in **Fig 8**. In the same way, BioSeq-Diabolo integrates the top 5 best predictors by using Learning to Rank. The integrated predictor is highly comparable with the other state-of-the-art predictors, including DIAMONDScore [13], DeepGO [35] and DeepGOCNN [53] (see **Tables** 4, **S4** and **S5**). These experimental results show that BioSeq-Diabolo is also able to facilitate the development of the computational predictors for protein function prediction.

## 4. Conclusion

Biological sequence similarity analysis plays a critical role in the biological structure and function studies. As the language of "the books of life", like natural language, biological sequence is

**Table 4. The performance of BioSeq-Diabolo compared with competing methods in Cellular Component Ontology (CCO) for protein function annotation.**

| Methods | AUPR[c] | Fmax[c] | Smin[c] |
|---|---|---|---|
| Naive[1] | 0.483 | 0.541 | 8.466 |
| DIAMONDScore[1] | 0.500 | 0.523 | 8.347 |
| DeepGO[1] | 0.446 | 0.503 | 5.791 |
| DeepGOCNN[1] | 0.523 | 0.582 | 5.234 |
| BioSeq-Diabolo[b] | 0.514 | 0.577 | 5.569 |

[a] The results of the competing methods were obtained from [54]. These competing methods and BioSeq-Diabolo were evaluated on the same test dataset. Therefore, these results can be directly compared

[b] The result of the best predictor constructed by BioSeq-Diabolo (integrating top 5 best predictor by using Learning to Rank). The input protein sequence embeddings are extracted by BioSeq-BLM [21] with Position-Specific method [55]. The input GO term embeddings are represented by label embedding matrix reported in [54]

[c] The performance evaluation indicators were described in **S2 Text** and details of the reported experiments were described in **S3** and **S4 Texts**.

information-complete, and has its own semantics. Based on biological language semantics, we can better understand the semantics of "the books of life". The platform BioSeq-Diabolo automatically analyses biological sequence similarities based on biological language semantic for different tasks in bioinformatics. Experimental results show that the predictors automatically generated by BioSeq-Diabolo even outperform the state-of-the-art predictors in the fields of protein remote homology detection, circRNA-disease association identification and protein function annotation, indicating that BioSeq-Diabolo will provide new concepts and techniques for biological sequence analysis, and facilitate the development of new computational predictors for biological sequence analysis. Although BioSeq-Diabolo incorporates some state-of-the-art biological sequence similarity analysis methods, we will focus on integrating more powerful algorithms into BioSeq-Diabolo in our future studies. We believe that BioSeq-Diabolo will play important roles in biological sequence analysis.

## Supporting information

**S1 Table. The distribution methods and their descriptions.**
(DOCX)

**S2 Table. Representation methods and their descriptions.**
(DOCX)

**S3 Table. Representation methods and their descriptions.**
(DOCX)

**S4 Table. The performance of BioSeq-Diabolo compared with competing methods in Molecular Function Ontology (MFO) for protein function annotation.**
(DOCX)

**S5 Table. The performance of BioSeq-Diabolo compared with competing methods in Biological Process Ontology (BPO) for protein function annotation.**
(DOCX)

**S1 Fig. The flowchart of BioSeq-Diabolo.** The web server and stand-alone package of BioSeq-Diabolo were developed based on this procedure.
(TIF)

**S2 Fig. The running time required for 9 different methods to get the result from scratch.** The running time contained training stage and test stage (50000 training examples and 50000 test samples). These experiments were performed on Intel(R) Xeon(R) CPU E5-2660 v3 (2.60 GHz with 10 cores) and memory of 64 G.
(TIF)

**S1 Text. Learning to Rank.**
(DOCX)

**S2 Text. Performance evaluation indicators.**
(DOCX)

**S3 Text. Method optimization and Result visualization.**
(DOCX)

**S4 Text. The details of the reported experiments.**
(DOCX)

## Author Contributions

**Conceptualization:** Bin Liu.

**Data curation:** Hongliang Li.

**Formal analysis:** Hongliang Li, Bin Liu.

**Funding acquisition:** Bin Liu.

**Investigation:** Hongliang Li, Bin Liu.

**Methodology:** Hongliang Li, Bin Liu.

**Project administration:** Bin Liu.

**Resources:** Bin Liu.

**Software:** Hongliang Li.

**Supervision:** Bin Liu.

**Validation:** Hongliang Li.

**Visualization:** Hongliang Li.

**Writing – original draft:** Hongliang Li.

**Writing – review & editing:** Hongliang Li, Bin Liu.

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
