## [Decision Letter · Decision Letter 0]

16 Dec 2022

Dear Prof. Liu,

Thank you very much for submitting your manuscript "BioSeq-Diabolo : biological sequence similarity analysis using Diabolo" for consideration at PLOS Computational Biology.

As with all papers reviewed by the journal, your manuscript was reviewed by members of the editorial board and by several independent reviewers. In light of the reviews (below this email), we would like to invite the resubmission of a significantly-revised version that takes into account the reviewers' comments.

As you will see from the reports below, the reviewers note the good results and usability of the code web server. However, they identify many significant issues. In particular, the manuscript lacks description of the methods (particularly of the use of LTR, the main contribution) and the evaluation experiments. Please ensure that the reader can evaluate from the text whether evaluation experiments support the desired conclusions, including a description of what information is held out from which parts of the approach to avoid data leakage and, more generally, what biological problem these experiments are simulating. Also, it seems that some performance numbers are copied from other papers; please reproduce the results or describe how you ensured that evaluation regime is exactly reproduced. Without substantial revisions such that the manuscript precisely describes its methods and accurate evaluation methodology, we will be unlikely to send the paper back to review.

We cannot make any decision about publication until we have seen the revised manuscript and your response to the reviewers' comments. Your revised manuscript is also likely to be sent to reviewers for further evaluation.

Sincerely,

Maxwell Wing Libbrecht, Ph.D.

Academic Editor

PLOS Computational Biology

Lucy Houghton

Staff

PLOS Computational Biology

As you will see from the reports below, the reviewers note the good results and usability of the code web server. However, they identify many significant issues. In particular, they manuscript lacks description of the methods (particularly of the use LTR, the main contribution) and the evaluation experiments. Please ensure that the reader can evaluate from the text whether evaluation experiments support the desired conclusions, including a description of what information is held out from which parts of the approach to avoid data leakage and, more generally, what biological problem these experiments are simulating. Also, it seems that some performance numbers are copied from other papers; please reproduce the results or describe how you ensured that evaluation regime is exactly reproduced. Without substantial revisions such that the manuscript precisely describes its methods and accurate evaluation methodology, we will be unlikely to send the paper back to review.

Reviewer's Responses to Questions

**Comments to the Authors:**

Reviewer #1: Li and Liu introduced a new method called BioSeq-Diabolo to calculate the biological sequence similarities. Biological sequence similarity analysis is an important task in bioinformatics, and efficient methods are desired. Generally, BioSeq-Diabolo is interesting, because it uses the techniques derived from the field of natural language processing, bringing new techniques and concepts to solve this task. The corresponding webserver has been constructed, and the standalone package has been released as well. I have tested both the webserver and the package, and they worked well as described. I believe that they will be particularly interesting for the researchers who are working on the related fields. I have the following comments to improve the presentation of this manuscript.

1. From the webserver, I understand that there are several pipeline software tools as listed in the right part. However, the relationships between these software tools and BioSeq-Diabolo should be explained in the webserver site as well. As a result, the users are able to use these software tools for their own tasks and aims.

2. The information or references of the datasets provided in the Download section (http://bliulab.net/BioSeq-Diabolo/download/) should be given.

3. I have downloaded the stand-alone package from http://bliulab.net/BioSeq-Diabolo/static/download/Sesica.tar.gz, and I felt this package is easy to use, especially for the large dataset analysis aims. However, I failed to find the files for the examples reported in the main text. I suggest Li and Liu providing this information so as to help the users reproducing the reported experiments.

4. The figure 2 is confusing. More information should be provided in its legend.

5. The structure shown in figure 4 is clear, which is very similar as the Diabolo, but the input and output shown in the top part flowchart is not clear. The two important components should be shown in different colors

6. In the current study, Li and Liu used four performance measures to evaluate their performance. Did they follow other studies, and why?

7. For a given methods in BioSeq-Diabolo, it is difficult for the users to select the best one for their own tasks. Did Li and Liu consider this problem? It will be more useful to add the function to automatically select the optimized methods for different tasks. Furthermore, there are several parameters for the different methods. The parameter optimization function should also be added.

8. The result visualization part is elegant and clear, but I cannot find the corresponding figures when using the standalone package. Can Li and Liu also add these visualization function into the standalone package as well?

9. This manuscript presented a very useful tool for biological sequence similarity analysis, but more discussions of its potential applications should be given in the conclusion section in order to attract the readers of Plos computational biology.

Reviewer #2: The authors established the web server and the stand-alone package for biological sequence similarity analysis called BioSeq-Diabolo. Its performance was well evaluated for different biological sequence similarity analysis problems. BioSeq-Diabolo is easily combined with their previously established software tools, and it is the first comprehensive platform incorporating various methods from natural language processing (NLP). It is reasonable to conduct the biological sequence similarity analysis based on the NLP methods considering the similarities between the biological sequences and the natural language sentences.

The following revisions are suggested:

1) The heterogeneous biological sequence similarity analysis shown in BioSeq-Diabolo will contribute to many problems in bioinformatics, such as non-coding RNA and disease association prediction. In the current version of the manuscript, it is not clear how to combine the heterogeneous features (such as non-coding RNAs and diseases) to make the prediction. More explanations are needed.

2) The information shown in table 1 clearly indicates that BioSeq-Diabolo and their previous method BLM are complementary, dealing with different problems. For example, Diabolo is for biological sequence similarity analysis, and BLM is for classification or sequence labelling problem in bioinformatics. More explanations of their relationships are need in both the main text and the web site of Diabolo.

3) Although figure 1 shows the similarities between sequence similarity analysis and semantics analysis, these similarities are not so clear. More explanations and discussions are needed, especially for their similarities among the individual steps shown in this figure.

4) In section 3.2, the concept of embedding is not clear. Dose it mean extract the features of the biological sequences?

5) Learning to Rank is a method to combine different methods. This method seems to be better than the other unsurprised ones. As discussed in the corresponding section, it has been successfully applied to solve many problems in bioinformatics. The authors are suggested to show more details of Learning to Rank, because it is an important step in Diabolo.

6) Written problems: the last column in Table 1, “Sequence similarities analysing” -> “Sequence similarity analysis”.

7) The web server of Diabolo is a big advantage, and it is easy to use. I only need to feed the features, and the web server will output the desired analysis results. I validated it with both the examples provided by the authors and my own data, and I got the desired results. So I think the web server will popular with the other users as well, but Some improvements are desired. For term explanation, the authors are suggested to add the corresponding references of the terms, which will help the user to get the correct information. The main buttons (“Server”, “Tutorial” and “Document”) are suggested to be more clear and bigger.

8) More discussions of the advantages and disadvantages of Diabolo are suggested to given in the conclusion part.

Reviewer #3: In this work, by using the LTR integration method to analyze the similarity of different biological sequences (DNA, RNA, protein, disease, etc), an unified platform for systematically analyzing the similarity of homogeneous and heterogeneous biological sequences has been realized. The text is logically coherent, and the server functions provided are complete, powerful and useful. Here are some comments and suggestions to further enhance this study.

Major

1) In the three tests, the descriptions of the evaluation indicators are missing, and they don't appear in the supplementary material either.

2) It is said that "The users only need to input the embeddings of the biological sequence data. BioSeq-Diabolo will intelligently identify the task, and then accurately analyse the biological sequence similarities based on biological language semantics." SO how BioSeq-Diabolo intelligently identifies tasks is not explained in the paper.

3) In the protein function annotation task, the official evaluation metrics in CAFA are F-max and S_min, which are missing in this work.

4. Lack of running time reports and performance evaluation of models under large-scale data. Estimation of time consumption is important for a user-oriented method.

Minor

1. Figure 1, a, the final protein should be "1D3B" instead of "D3B"

2. Figure 4, "Learning to Rank" part, "Predicted diseases list for topic", "topic" is used in NLP background. Here it should be "Predicted diseases list for RNAs". The same problem exists in the Figure of the server(http://bliulab.net/BioSeq-Diabolo/server/).

**Have the authors made all data and (if applicable) computational code underlying the findings in their manuscript fully available?**

Reviewer #1: None

Reviewer #2: Yes

Reviewer #3: Yes

PLOS authors have the option to publish the peer review history of their article (what does this mean?). If published, this will include your full peer review and any attached files.

Reviewer #1: No

Reviewer #2: No

Reviewer #3: No
---

## [Editor Report · Decision Letter 1]

16 Mar 2023

Dear Prof. Liu,

Thank you very much for submitting your manuscript "BioSeq-Diabolo : biological sequence similarity analysis using Diabolo" for consideration at PLOS Computational Biology.

As with all papers reviewed by the journal, your manuscript was reviewed by members of the editorial board. We would like to invite the resubmission of a significantly-revised version that takes into account the comments below.

Thank you for your revised manuscript. Upon initial reading, it seems that a key comment from the reviewers (e.g. R3-C1) remains unaddressed. It seems that descriptions of AUC and AUPR in general were added, but there is still no description of the experiments themselves. Please ensure that the reader can fully understand and reproduce each experiment from the text (main or supplementary), including the source of the data, all preprocessing performed, how fields from the source database were mapped to machine learning concepts (e.g. how true label y_i is defined for each task), the division of data into train and test sets, etc. This issue must be addressed before the manuscript can be re-sent for review. Also, when referencing the supplemental material, please include a reference to the specific section in question.

We cannot make any decision about publication until we have seen the revised manuscript and your response to the reviewers' comments. Your revised manuscript is also likely to be sent to reviewers for further evaluation.

Sincerely,

Maxwell Wing Libbrecht, Ph.D.

Academic Editor

PLOS Computational Biology

Lucy Houghton

Staff

PLOS Computational Biology

Thank you for your revised manuscript. Upon initial reading, it seems that a key comment from the reviewers (e.g. R3-C1) remains unaddressed. It seems that descriptions of AUC and AUPR in general were added, but there is still no description of the experiments themselves. Please ensure that the reader can fully understand and reproduce each experiment from the text (main or supplementary), including the source of the data, all preprocessing performed, how fields from the source database were mapped to machine learning concepts (e.g. how true label y_i is defined for each task), the division of data into train and test sets, etc. This issue must be addressed before the manuscript can be re-sent for review. Also, when referencing the supplemental material, please include a reference the specific section in question.
---

## [Decision Letter · Decision Letter 2]

18 Apr 2023

Dear Prof. Liu,

Thank you very much for submitting your manuscript "BioSeq-Diabolo : biological sequence similarity analysis using Diabolo" for consideration at PLOS Computational Biology. As with all papers reviewed by the journal, your manuscript was reviewed by members of the editorial board and by several independent reviewers. The reviewers appreciated the attention to an important topic. Based on the reviews, we are likely to accept this manuscript for publication, providing that you modify the manuscript according to the review recommendations.

Sincerely,

Maxwell Wing Libbrecht, Ph.D.

Academic Editor

PLOS Computational Biology

Lucy Houghton

Staff

PLOS Computational Biology

Reviewer's Responses to Questions

**Comments to the Authors:**

Reviewer #1: It is revised well

Reviewer #2: The authors have improved their work and addressed my comments. Happy to recommend this version for publication in PLoS CB.

Reviewer #3: The author has actively responded to our comments and resolved most of them. Here are some additional suggestions to further enhance this study.

1. The author added "The details of the reported experiments" section in the supplementary material to further explain the details of the experiment. But as far as the protein function annotation problem is concerned, the experimental implementation has the following problem:

Generally speaking, when evaluating protein function annotation, most of the researchers will evaluate MFO, BPO, and CCO separately, instead of only evaluating CCO.

2. I think the training time should not be the focus of the evaluation. Instead, the experiment should focus on evaluating the time consumption between different approaches. That is, when the number of samples is the same, the time required for different methods to get the result from scratch.

3. "BioSeq- Diabolo will intelligently identify the specific tasks (homogeneous or heterogeneous biological sequence similarity analysis tasks) based on the input embeddings of the biological sequences and the parameters provided by the users." This section needs to specify how to identify specific tasks.

4. In addition, users only need to input biological sequence data, but generating different input embeddings from biological sequence data requires different models. How does BioSeq-Diabolo select the corresponding model.

5. The layout of Figure 4 is suitable, and many subplots are too small to be seen clearly. You can try to regroup the subplots. The theme of Diabolo does not need to be presented on this diagram, which hinders the normal layout of the diagram and can be reflected in your LOGO.

**Have the authors made all data and (if applicable) computational code underlying the findings in their manuscript fully available?**

Reviewer #1: None

Reviewer #2: Yes

Reviewer #3: Yes

PLOS authors have the option to publish the peer review history of their article (what does this mean?). If published, this will include your full peer review and any attached files.

Reviewer #1: No

Reviewer #2: No

Reviewer #3: No

Figure Files:

Data Requirements:

Reproducibility:

References:

---

## [Editor Report · Decision Letter 3]

24 May 2023

Dear Prof. Liu,

We are pleased to inform you that your manuscript 'BioSeq-Diabolo : biological sequence similarity analysis using Diabolo' has been provisionally accepted for publication in PLOS Computational Biology.

Best regards,

Maxwell Wing Libbrecht, Ph.D.

Academic Editor

PLOS Computational Biology

Lucy Houghton

Staff

PLOS Computational Biology

---

## [Editor Report · Acceptance letter]

12 Jun 2023

PCOMPBIOL-D-22-01464R3 

BioSeq-Diabolo : biological sequence similarity analysis using Diabolo

Dear Dr Liu,

I am pleased to inform you that your manuscript has been formally accepted for publication in PLOS Computational Biology. Your manuscript is now with our production department and you will be notified of the publication date in due course.

With kind regards,

Zsofi Zombor
